# Enhancing Mobile Edge Computing with Efficient Load Balancing Using Load Estimation in Ultra-Dense Network

**DOI:** 10.3390/s21093135

**Published:** 2021-04-30

**Authors:** Wen Chen, Yongqi Zhu, Jiawei Liu, Yuhu Chen

**Affiliations:** School of Information Science and Technology, Donghua University, Shanghai 201620, China; chenwen@dhu.edu.cn (W.C.); 2201777@mail.dhu.edu.cn (J.L.); 2201780@mail.dhu.edu.cn (Y.C.)

**Keywords:** software defined network (SDN), ultra dense network (UDN), mobile edge computing (MEC), load balancing, subtask, genetic algorithm (GA), ping-pong effect

## Abstract

With the exponential growth of mobile devices and the emergence of computationally intensive and delay-sensitive tasks, the enormous demand for data and computing resources has become a big challenge. Fortunately, the combination of mobile edge computing (MEC) and ultra-dense network (UDN) is considered to be an effective way to solve these challenges. Due to the highly dynamic mobility of mobile devices and the randomness of the work requests, the load imbalance between MEC servers will affect the performance of the entire network. In this paper, the software defined network (SDN) is applied to the task allocation in the MEC scenario of UDN, which is based on routing of corresponding information between MEC servers. Secondly, a new load balancing algorithm based on load estimation by user load prediction is proposed to solve the NP-hard problem in task offloading. Furthermore, a genetic algorithm (GA) is used to prove the effectiveness and rapidity of the algorithm. At present, if the load balancing algorithm only depends on the actual load of each MEC, it usually leads to ping-pong effect. It is worth mentioning that our method can effectively reduce the impact of ping-pong effect. In addition, this paper also discusses the subtask offloading problem of divisible tasks and the corresponding solutions. At last, simulation results demonstrate the efficiency of our method in balancing load among MEC servers and its ability to optimize systematic stability.

## 1. Introduction

With the development of 5G technology and the Internet of Things (IoT), the number of wireless devices is exploding exponentially and the application scenarios of IoT are becoming more and more diversified. For example, there are a large number of computationally intensive and delay-sensitive applications, such as virtual reality, online games, and so on, which need strong computing power to meet the requirements of ultra-low latency. In order to meet this demand, in recent years, the traditional cloud computing network architecture is gradually turning to the MEC network, and the computing services and functions originally located in the core cloud data center are gradually moving towards the edge of the network. Through the MEC network, users can be provided with ubiquitous computing, storage, communication and other services through base stations and wireless access points closer to them, so as to effectively reduce the computing delay and energy consumption, and greatly improve the resource utilization of the whole network [1,2,3].

UDN solves more wireless access requirements by deploying dense base stations in the community so as to provide user equipment (UE) with huge access capacity and improve the capacity of the whole network. Hence, the combination of UDN and MEC cannot only provides services for more UE but also meets the computing power and latency requirements of UE for some specific tasks at the same time [4,5,6].

Even though the combination of UDN and MEC can bring great advantages and benefits, there are inevitably some problems to be solved, such as load balancing between MEC servers, which is mainly caused by the highly dynamic distribution of UE temporally and spatially and the randomness of UE application tasks [7]. Excessive offloading tasks to the same MEC server will lead to server overload and congestion, which will not only impact system performance and server life, but also lead to a sharp decline in term of quality of service (QoS) and quality of experience (QoE) [8,9,10]. On the contrary, it is a huge waste of computing resources when servers running under low load [11]. Consequently, it is urgent to solve the problem so as to achieve load balancing between the MEC servers in UDN.

SDN is a new network architecture commonly associated with the OpenFlow protocol. With Openflow, the control of network traffic can be realized flexibly by separating the control plane of network devices from the data plane. Based on the advantage of SDN, the SDN paradigm can be applied to the task assignment in MEC. The MEC servers route the corresponding information to the controller, and then the controller sends back the command according to the scheme.

The UE’s task offloading scheme greatly affects the load balance between MEC servers. It determines which MEC server to assign processing tasks to. To achieve load balancing between servers, we need to design an appropriate global offloading scheme for all tasks applied by UE during each decision period. Moreover, it is an NP-hard problem that becomes more and more complex as the number of tasks increases. At present, there are three main designs of MEC offloading strategies: (1) Convex optimization theory is used to design the optimal or suboptimal computing offloading strategy. However, this process may take a long time, which is contrary to the original intention of MEC to shorten the computing delay. (2) A heuristic algorithm is used to design offloading strategies with low complexity. However, this method often lacks theoretical support and cannot achieve better computing performance. (3) Artificial intelligence technology is used to design fast and efficient offloading strategy. However, it relies heavily on historical data. In addition, due to the highly dynamic MEC environment, the correlation between training data and real-time data is low [7]. Therefore, this paper innovatively proposes a fast and effective load balancing algorithm to solve the above problems.

According to the characteristics of UE’s tasks, they can be divided into several different subtasks. According to the interdependence among subtasks, there are three main types of computing tasks, i.e., sequential task, parallel task, and sequential-parallel-hybrid task [12]. By dividing the task into subtasks, the resource utilization can be improved, and the calculation delay and energy consumption of the system can be further reduced. However, it is difficult to take this into account in most of the work on load balancing.

The main contributions of this paper are as follows:By incorporating the user load prediction into the load-balancing scheme, a ping-pong effect solution based on load estimation is proposed. Due to the highly dynamic environment of MEC, if the load strategy considers the currently applied tasks and the load of each MEC server only, it will cause too much unnecessary task transmission between MEC servers. Meanwhile, it will result in unexpected load overhead, which is called the ping-pong effect in load balancing. In fact, the impact of the ping-pong effect can be reduced by our approach.A new concept is proposed: *task unit load transfer overhead*. On this basis, we propose a low complexity and effective load balancing algorithm. In fact, the simulation results also show the effectiveness of the algorithm. Compared with the genetic algorithm (GA), our algorithm has a better time-consuming performance.In the case of divisible tasks, the load of each MEC on the timeline is first sensed, and then a balancing decision based on the interdependency of subtasks is made. In particular, for sequential-parallel-hybrid tasks, the hierarchical method of subtasks is used to simplify and specify the relationship between them. Furthermore, the effectiveness of the above methods is verified through experimental results.

The remainder of this paper is arranged as follows. In Section 2, some related work is introduced generally. In Section 3, the system model consisting of system framework, task model, sub-task model, and load estimation is proposed in detail. In Section 4, solutions and corresponding algorithms to solve the problem of load balancing are proposed. In Section 5, simulation results are discussed. Finally, in Section 6, some conclusions of the whole paper are summarized.

## 2. Related Work

It is of great significance to study MEC under UDN. A lot of research work has been carried out in recent years.

Reference [13] studied how to choose the MEC server that users can directly connect to offload tasks to minimize the cost-effectiveness of operators and users and compared the total delay and resource utilization of Q-learning, DQN, and game theory. Reference [8] proposes a heuristic greedy offloading scheme to solve the offloading problem of MEC. However, it only considers the total delay of the entire system but ignores the delay requirement of each task. Reference [4] formulated task-offloading as an integer nonlinear programming problem and proposed an efficient task-splitting and channel resource allocation scheme based on a differential evolution algorithm. It can obviously reduce energy consumption and has good convergence. Nevertheless, the computing resources allocated to each task are the same, and the delay requirements of the tasks were not taken into account; thus the resources might not be fully utilized. Reference [14] proposes an iterative algorithm based on the Successive Convex Approximation (SCA) method to minimize the energy consumption of the equipment and meet the delay requirement of the task. Reference [15] proposed a user-centric energy-aware mobility management scheme, which optimizes the delay caused by radio access and computation under the constraints of long-term energy consumption of users. All these efforts are focused on solving the system energy consumption and delay while ignoring the impact and consequences of the unbalanced load of the MEC servers.

Considering the dynamic arrival of tasks and limited computing resources, Reference [9] proposed an online computing offload mechanism to optimize the system performance and achieve sub-optimal performance. Reference [16] proposes an online peer-to-peer offloading framework for small cell base stations, which uses Lyapunov technology to maximize long-term system performance while keeping the energy consumption of small base stations under various long-term constraints. Reference [17] considered the three-tier collaborative computing network of devices, edge nodes, and cloud servers and optimized the framework by using the alternating direction method of multipliers (ADMM) and the difference in convex functions (DC), so as to minimize the average task duration when the equipment power is limited. None of the above work considers the delay limit of the task. Although load balancing is performed on the MEC servers, they only focus on short-term load balancing, which will lead to a ping-pong effect and cannot achieve long-term load balancing. References [7,10] introduced the idea of prediction in load balancing. The specific method is such that the size and processing time of UE application tasks follow an exponential distribution. Hence, after knowing the number of users covered by each small base station, their approximate load can be sensed. In [18], judging from the trajectory of Google cluster, the arrival time and service time of job requests are exponential distributed can be regarded as a Poisson process. It is believed that the arrival process of the work request is not a stationary process, but is essentially a pseudo-stationery process [19], and it can use the load of the previous moment to predict the load of the next moment. However, load forecasting based on the law of large numbers may not be ideal. Reference [20] mentioned that due to work and study, more users often gather in certain fixed areas, while fewer users are located in other areas. As a result, it is easy to cause an unbalanced distribution of cellular traffic. Reference [21] mentions that the traffic fluctuation of small cells is much greater than that of large ones since small cells and cannot benefit from the law of large numbers. Reference [22] proposed that global popularity can only represent the average content request trend of many users. Therefore, it is more reasonable to use user preferences than global popularity to maximize the click-through rate. From the above work, in order to make the load prediction more accurate and reasonable, we can consider the load prediction for each user.

## 3. System Model

In this section, we propose the system framework, the task model, and the load estimation.

### 3.1. System Framework

As shown in Figure 1, this paper presents a system framework that is a network scenario of MEC in 5G UDN. In a macro cell, a macro base station (MBS) is deployed in its center. Under the coverage of an MBS, *B* small base stations (SBSs) are placed. A large amount of UE is randomly distributed near the SBS to request services through a wireless link. We denote the set of SBSs as B={1,2,...,B} and the set of UEs covered by the m${}_{th}$ SBS ∈B as $U_{m}=\left\{1,2,...,U\right\}$. Each base station is equipped with an MEC server to process the tasks offloaded by UEs. Adjacent MEC servers are all connected by fiber links to transmit data [4,16,17,23]. In UDN, this is feasible due to the short distance between SBSs. The fiber link provides a stable, high-speed, and reliable connection, saving channel resources for the communication between the UEs and the SBS.

SDN plays an important role in the system model. First, it hierarchizes the network devices in the system and defines different levels with different functions. Second, it monitors the system environment changes through network devices to obtain real-time environment variable parameters. Third, it runs the user task algorithm according to these parameters and gets the task offloading instruction of each task. The instruction indicates which MEC server the task is executed by. The specific content of SDN is as follows.

In order to apply the SDN paradigm to the above architecture, we introduced the core ideas of SDN under different circumstances from the normal SDN scenario. The main content of SDN consists of two parts: (1) control plane and (2) data plane. Compared with the traditional network, it separates the controller from the data plane. On this basis, we let MBS and its MEC server represent the SDN controller and let all the SBSs and their MEC servers represent the data plane. However, the MEC scenario of UDN is different from the traditional data network. Its responsibility is not only to forward packets of network elements. Therefore, the functions proposed by traditional SDN are not sufficient in this case. In order to make it suitable for this scenario, its concept needs to be expanded appropriately. In the traditional SDN framework, the network elements in the data plane are limited to forwarding data packets, and the controller determines the forwarding rules. In addition, we also need a data plane to obtain a global view of network information, including user distribution, MEC server load status,  task request information, etc. In more detail, if the UE is within the coverage of this SBS, the UE distribution can be obtained through the wireless link between the UE and the SBS. The MEC server can obtain its load status, including current processing progress of these tasks and the computing resources spent by the UE tasks. If the UE has a task request for processing, it will send the task information to the SBS before the task starts running. In each decision slot, each SBS sends this information to the MBS. Then, the controller aggregates the messages from all SBSs and makes decisions through load balancing algorithms. Specifically, the offloading decision determines which MEC server the user’s task will be offloaded to, so as to balance the load among the MEC servers while meeting the task delay requirements.

### 3.2. Task Model

#### 3.2.1. Task Information

If the UE applies for a service from the MEC server that has established a link connection, the UE will upload a delay-sensitive task to the SBS. The task has three parameters that can be denoted as T={s,c,d}, and then the task information of $n_{th}$ UE $\in U_{m}$ represents $T_{n,m}=\left\{s_{n,m},c_{n,m},d_{n,m}\right\}$. The details of T are as follows [24,25]:*s* indicates the size of task data. If the algorithm needs to transfer a task from one MEC server to another MEC server through an fiber link, it will affect the task transmission delay.*c* represents the total number of CPU cycles required to complete the task. The MEC server will allocate some computing resources (*cycles/s*) to process these CPU cycles if the task is selected to be executed in it. Then, the computing resource used to perform the offloading task becomes the MEC server load.*d* represents the maximum time delay that can be tolerated to complete the task, that is, the time from when the task is offloaded to the end of the task.

These parameters are related to the specific type and requirements of computing tasks and can be obtained through task analysis and modeling [26,27,28].

#### 3.2.2. Task Offloading

Similarly to other studies in the literature [7,9,29], this paper mainly studies the computational load balancing of MEC servers in UDN but does not consider the communication model of task upload. In other words, its purpose is to balance the load of tasks that have reached the MEC server but have not yet begun to execute. Another reason is that although most works, for example, [16,17], introduce a communication model, the allocated channel bandwidth, upload power, and signal interference of each UE are considered fixed. According to the Shannon equation, the upload rate of the UE can be easily obtained as a fixed value, resulting in a fixed upload delay of the task. In this case, there is no need to introduce a communication model. If the above parameters are treated as dynamically variable, it will make the entire system model too complicated and difficult to solve.

Since we did not consider the task upload process in the model, we need to make appropriate changes in *d* of T to fit our model. We denote *d* as the maximum tolerance delay of the task minus the transmission delay of the task upload. That is to say, *d* is the remaining time to complete the task within the maximum delay range after the task is uploaded.

There are two methods of task offloading. One is to offload in the initial MEC server (IMEC), which has established a link with the UE. The other is to offload in non-initial MEC server (NMEC), and the task is transferred from the initial MEC server to the server.

We can get a basic equation when a task is computing in one MEC server, expressed as
(1)c=t·l,
where *t* denotes the processing time in MEC server and *l* is the computing resources (in CPU cycles/s) allocated by the MEC server.

(1) *Offloading in IMEC:*

We let $l_{n,m}^{I}$ represent the task computational load of $n_{th}$ UE $\in U_{m}$. When the task is processed in it, it is calculated into the load of the IMEC server.

If the task is completed at the maximum tolerance delay $d_{n,m}$, we obtain the following equation from Equation (Equation 1):(2)$$c_{n,m}=d_{n,m}\cdot l_{n,m}^{I},$$
then, the $l_{n,m}^{I}$ can be expressed as:(3)$$l_{n,m}^{I}=\frac{c_{n,m}}{d_{n,m}}.$$

(2) *Offloading in NMEC:*

The transfer of the task from IMEC to NMEC will cause a certain transmission delay, and it can be expressed as
(4)$$t_{n,m}^{trans}=\frac{s_{n,m}}{r^{trans}},$$
where $r^{trans}$ is the transmission rate of optical fiber. Then, the processing time for completing the task at the maximum tolerance delay in NMEC can be represented as
(5)$$t=d_{n,m}-t_{n,m}^{trans}.$$

After the completion of the task, there will be a process of returning the calculation results. Since the calculation results are usually very small and the transmission speed of optical fiber is fast and stable, we ignore the return delay.

Since the transmission delay takes up part of the time $d_{n,m}$, it is initially used for task processing, compared with offloading in IMEC. It will not complete the task within the maximum delay if it still allocates the same computing resources as $l_{n,m}^{I}$. Therefore, it is necessary to increase the allocation of computing resources to meet the delay demand. Furthermore, let this part of the increased computing resources be denoted as $l_{n,m}^{Extra}$, which is defined as *task transfer load overhead* (TTLO). Then, when the task is processed in the NMEC server, the task calculation load $l_{n,m}^{N}$ of $n_{th}$ UE $\in U_{m}$ can be expressed as follows:(6)$$l_{n,m}^{N}=l_{n,m}^{I}+l_{n,m}^{Extra}.$$

We bring Equations (Equation 5) and (Equation 6) into Equation (Equation 1); then, we can get
(7)$$c_{n,m}=\left(d_{n,m}-t_{n,m}^{trans}\right)\cdot \left(l_{n,m}^{I}+l_{n,m}^{Extra}\right).$$

Further, $l_{n,m}^{Extra}$ can be given as follows from Equation (Equation 7):(8)$$l_{n,m}^{Extra}=l_{n,m}^{I}\cdot \frac{t_{n,m}^{trans}}{d_{n,m}-t_{n,m}^{trans}}.$$

#### 3.2.3. Subtask Model

In order to complete a computationally arduous task efficiently and improve the utilization of computing resources, the task can be divided into several related subtasks according to the characteristics of the task. Due to the different ways of dividing subtasks, the running order, and relationship between subtasks will change, there are three types of subtasks according to the interdependency among them, which are sequential subtasks, parallel subtasks, and sequential-parallel-hybrid subtasks. We assume that after the entire task is divided, we know the task information set T of each subtask, because we know the detailed division process.

(1) *Parallel subtasks:*

There are no dependencies between subtasks. Therefore, they are considered as independent subtasks that can be executed in parallel; e.g., some video tasks can be divided into frames and processed simultaneously. Subtasks only need to be completed within their respective delay deadlines to ensure the delay requirements of the overall task.

(2) *Sequential subtasks:*

Subtasks need to be executed one by one; e.g., face recognition includes three sequential subtasks: face detection, image processing feature extraction, and face recognition. After each subtask is computed completely, it will generate a certain amount of calculation results. The latter subtask must receive the calculation results of the previous subtask before it can be executed. Hence, the start time of execution of the latter subtask is the end time of execution of the previous subtask. The whole process does not finish until the last subtask obtains the final calculation result. For example, in Figure 2a, $t_{n}$ represents the task processing time of *subtask n*, i.e., the maximum tolerance delay *d*, and *subtask2* can only be executed after $t_{1}$ seconds, etc., until *subtask4* gets the final result after execution. If all subtasks are not executed in the same MEC server, the calculation results between the two subtasks will be transmitted between MEC servers via optical fiber. Due to the small scale of data, we still ignore the delay of data transmission.

(3) *Sequential-parallel-hybrid subtasks:*

Because of the coexistence of sequential and parallel relationships among subtasks, the execution order and execution time of each subtask become complicated. In order to simplify and refine them, we propose a hierarchical method of sub-tasks. Specifically, we divide all subtasks into several layers. The subtasks in the same layer can be processed in parallel, and the subtasks between different layers need to be executed in order. The layer of a subtask depends on the maximum number of layers of its previous subtasks. For example, as shown in Figure 2b, *subtask1*, *subtask2*, and *subtask3* can be computed in parallel because of the same layer they are at. *subtask4* must receive the results of both *subtask1* and *subtask2* before it can be computed. It is worth noting that $t_{1}$ and $t_{2}$ may not be equal; therefore, the execution time of *subtask4* depends on the larger one. The previous subtasks of *subtask6* are *subtask1* at *layer1* and *subtask4* and *subtask5* at *layer2*. We place it on *layer3* because the largest layer of its previous subtasks is *layer2* even though *subtask1* is at *layer1*.

### 3.3. Load Estimation

In this section, we introduce the idea of load estimation into our model. Load prediction allows each SBS to estimate its approximate load status. As a result, the SDN controller can learn the global load status and make corresponding load balancing decisions. The reasons for introducing load prediction are explained by the following situations that may take place in the MEC network.

#### 3.3.1. Ping-Pong Effect

Due to the user’s temporal and spatial mobility and the random arrival of UE tasks, the environment of the MEC system is highly dynamic. Therefore, the load of each MEC server also changes from time to time. Our goal is to maintain the load balance of all MEC servers for a long time and offload part of the load from high-load MEC servers to low-load MEC servers. However, UEs covered by the a certain SBS do not always apply for task service at the same time, which will lead to the following two cases occurring.

Since we are trying to consider the whole situation more comprehensively, when all UEs offload their tasks at the same time, we let the real high-load MEC (RHL-MEC) and real low-load MEC (RLL-MEC) represent the high-load MEC server and low-load MEC server, respectively. Another explanation for the classification is that we consider the load of all UEs covered by each SBS, regardless of whether the UE has offloaded the task. In order to achieve it, estimated load is then proposed in Section 4.1.2. Let ε represent the load difference between RHL-MEC and RLL-MEC at this time. In the case of ignoring the TTLO, as shown in Equation (Equation 8), RHL-MEC offloading $\frac{\varepsilon }{2}$ to RLL-MEC can balance the load. We assume that all UEs in Figure 3 have the same task load requirements, so that we can explain the following situations in this section in this section. In this case, we can divide the MEC servers into RHL-MEC and RLL-MEC according to the number of UEs. In addition to this assumption, the classification needs to be based on the estimated load of all MEC servers.


*Case 1: Incorrect direction of load balancing:*


The load of RHL-MEC may be less than that of RLL-MEC at some point; then, part of the load of RLL-MEC will be offloaded to RHL-MEC for load balancing. As shown in Figure 3, as far as *SBS1* and *SBS2* are concerned, at a certain moment, all five UEs of *SBS2* apply for task processing, and the number of UEs of *SBS1* is less than this number, even though its total number of UE is seven. In order to balance the load between them, part of the load of *SBS2* needs to be offloaded to *SBS1*.


*Case 2: Excessive load balancing:*


The load of RHL-MEC exceeding RLL-MEC may be greater than ε at some point; then, the load exceeding $\frac{\varepsilon }{2}$ will be offloaded from RHL-MEC to RLL-MEC for load balancing. For example, seven UEs of *SBS1* all apply for task processing, and only one task of five UEs of *SBS2* is applied for processing; then, three tasks can be transferred from *SBS1* to *SBS2* to computed for load balancing.

If the load balancing decision only depends on the actual load of the MEC server at a certain time, the above two situations will occur randomly in the MEC network, which will cause tasks to be transferred back and forth between RHL-MEC and RLL-MEC. It is the so-called ping-pong effect of load balancing in the MEC network that may lead to unexpected TTLO. The reason for this effect is that the MEC server cannot estimate its approximate load limit, so the controller cannot grasp the global load status. Once the MEC server takes the estimate, we can get the correct direction of load transfer in load balancing and the specific amount of load that needs to be transferred. For example, in Figure 3, only the tasks of *SBS1* need to be transferred to *SBS2*, and only one task needs to be transferred.

Let us give an example to describe the ping-pong effect in detail:

(1) When t=0, all five UEs of *SBS2* apply for task processing, and the number of UEs that *SBS1* applies for task is three. At this time, the remaining four UEs of *SBS1* do not apply for task processing. If we only consider the load that has been applied to the task processed at this time to balance the load, we need to transfer the task of one of the five UEs to *SBS1*. Then, the load status of *SBS1* and *SBS2* is [*SBS1*: 4, *SBS2*: 4], where the number 4 represents the number of tasks loaded at this time.

(2) When $t=t_{1}$, where $t_{1}$ is the next decision slot, the remaining four UEs of *SBS1* apply for task processing. Taking into account the time cost of task completion and the continuity of task processing, the decision time slot is generally less than the task-processing time. The tasks offloaded in the previous step (t=0) may not be completed. We assume that none of the eight tasks have been completed. In this case [*SBS1*: 8, *SBS2*: 4], two tasks of the remaining four UEs of *SBS1* need to be transferred to *SBS2* to balance the load. Therefore, the load status of *SBS1* and *SBS2* is [*SBS1*: 6, *SBS2*: 6].

After the above two steps, the total number of task transfers is 3. In fact, if we take the load of the remaining four UEs into account, only one task needs to be transferred from *SBS1* to *SBS2* to balance the load ([*SBS1*: 7, *SBS2*: 5] → [*SBS1*: 6, *SBS2*: 6]). This is just a simple example; the actual situation will be more complex and changeable than the case above, which will lead to a more serious ping-pong effect.

#### 3.3.2. User Load Prediction

User load prediction can be used to estimate the load of an MEC server, which has been studied in some works using the law of large numbers. However, it may not be suitable for a small unit due to the user’s personal behavior preferences and task load requirements. For example, in Figure 3, the number of UEs covered by *SBS1* is more than that of *SBS2*, but it does not mean that the load of *SBS1* must be higher than that of *SBS2*. It is possible that users of *SBS2* tend to apply for tasks that are computationally intensive, while users of *SBS1* tend to apply for lighter tasks. In this case, the load of *SBS2* may exceed that of *SBS1*. For this reason, in order to make the load estimation more accurate, we carry out the load prediction for each user instead of using the law of large numbers.

In [30,31], the time-series analysis is used to predict user behavior, which can also be utilized to predict user load. Specifically, a cycle time is subdivided into several time periods, from which the user load is predicted. For example, a week can be divided into seven specific days, because the behavior and needs of users may be different on each day. Moreover, to make the load prediction more accurate, we can further analyze three time periods of a day: morning, afternoon, and evening. Since the task load of different application programs may vary, it is necessary to distinguish the usage of them by users. In general, we record the user’s task load requirements for different application programs in each time segment and then analyze the user’s approximate load requirements at this time.

Nonetheless, the above is just a simple mention of how to do load prediction. The specific load-prediction algorithm is beyond the scope of this paper.

#### 3.3.3. Overlapping Coverage

Due to the short distance between SBS, a UE may be covered by several SBSs. It will lead to a wrong balancing strategy when we rely on the UEs covered by a single SBS but ignore the overlapping coverage. As shown in Figure 3, we also assume that all UEs have the same task load requirements and that each MEC server has the same computing capacity in this section.

In terms of the number of UEs, *SBS1* and *SBS3* are both greater than *SBS2*, but this does not mean that the load of *SBS1* and *SBS3* is higher than that of *SBS2*. Although the computing capacity doubles when *SBS1* is added up to *SBS3*, they only need to serve nine UEs. This means that an MEC server serves 4.5 UEs that are less than the five UEs of *SBS2*. Therefore, in order to make the load estimation more reasonable in the case of multiple coverages, it is necessary to average the load of the UE under multiple coverages. For example, in Figure 3, the load of the four UEs covered by *SBS1* and *SBS3* needs to be evenly distributed to them, which is equivalent to the fact that *SBS1* covers five UEs and *SBS3* covers four UEs. In this way, we can get a more reasonable load state, such that the load of the MEC server of *SBS2* is actually the highest of the three in the network. Based on this, we can make a more correct load balancing decision.

## 4. Problem Solving

### 4.1. The Load of MEC Server Formulation

In this section, the estimated load of an MEC server is introduced to solve the ping-pong effect in load balancing. Furthermore, we then provide solutions to solve the load balancing problem of subtasks in MEC network.

#### 4.1.1. Load Timeline for Subtask

From Section 3.2.3, we conclude that after task division, due to the interdependency among subtasks, the execution time of the subtasks does not always start from the current moment. Consequently, if we still consider the load balancing strategy of these subtasks according to the current load and determine which MEC server the subtasks processed on, it will cause a certain load fluctuation that affects load balancing.

In order to solve the above problem, a timeline for the load of each MEC server is established to estimate the load variation from current moment. It is feasible since we can obtain the minimum delay requirement of each task and subtask and accordingly allocate computing resources to them. Then, we can obtain the execution completion time of a task when it is offloaded. Specifically, when making a load balancing decision for a subtask, we consider the load of all MEC servers at the execution start time on the timeline. For example, in Figure 2a, *subtask3* should consider the load in $t_{1}+t_{2}$ time from the current moment. Similarly, in Figure 2b, *subtask6* should consider the load in t time from current moment, which is the maximum of $t_{1}+t_{4}$ and $t_{2}+t_{4}$.

#### 4.1.2. The Estimated Load of MEC Server

In order to reduce the impact of the ping-pong effect in load balancing, instead of directly depending on their current load to make load balancing decisions, the maximum load of each MEC server should be estimated before that. In each time slot, not all UEs are covered by an SBS receive task processing services from the MEC server. The purpose of load estimation is to consider the load of UEs without offloading tasks into consideration in load balancing decision, and the method is user load prediction mentioned in Section 3.3.2.

Then, as shown in Figure 4a, the estimated load at the current time consists of two parts, where one is occupied by the task in progress, and the other is the predicted load of the UEs without offloading tasks. In addition to the two parts, the estimated load of other time points (t > 0) on the timeline may also include the load of the subtasks executed currently. It is worth noting that the green load will gradually decrease on the timeline as some tasks may be completed. As for the orange load, it may increase or decrease, depending on whether the task is offloaded in IMEC or NMEC. If the task is offloaded in NMEC, it decreases. Furthermore, if the task is offloaded in IMEC, the corresponding green load will be replaced with the predicted load of the user when it is completed.

In Figure 4b, load estimation and timeline are introduced in more detail. At the current time, the tasks of *UE1*, *UE2*, and *UE6* are being processed, where the task of *UE6* is transferred from another MEC server. The load of these three tasks is marked in green. Since *UE3* and *UE4* do not offload tasks, their predicted load is marked in orange. *UE5* has two sequential subtasks to process, in which *subtask2* starts to be processed after *subtask1* is completed. That is, *subtask1* is completed at $t_{2}$ time, while *subtask2* is completed at $t_{3}$ time. Therefore, on the timeline, the brown load represents the load of *subtask1* from current time to time $t_{2}$ and the load of *subtask2* from time $t_{2}$ to time $t_{3}$. After *subtask2* of *UE5* is completed, *UE5* has no task to process. Then, starting from time $t_{3}$, its predicted load is marked in orange and positioned on the timeline. Moreover, at time $t_{1}$, the tasks of *UE2* and *UE6* are completed. Then, the green load of *UE2* is replaced by its predicted load in orange. However, the green load of *UE6* has been removed, because *UE6* is not covered by the SBS. Furthermore, its predicted load in orange should be added to its IMEC.

### 4.2. Problem Formulation

In this section, we will detail the problem formulation process.

As shown in Figure 4a, through load estimation, we can obtain the approximate maximum load of each MEC server in UDN. Furthermore, then, a *mean load line* is obtained, which indicates the expected ideal load of each MEC server through the load balancing decision, under the condition that all UEs offload their tasks. The MEC server whose estimated load is above the *mean load line* is RHL-MEC, such as *MEC2*, *MEC4*, *MEC5* in Figure 4a. Otherwise it is RLL-MEC, such as *MEC1*, *MEC3*. Moreover, through load estimation, one can be made aware of how much load needs to be transmitted away from each RHL-MEC to the *mean load line*, and on the contrary, and how much load should be transmitted into each RLL-MEC to reach the *mean load line*.

Due to the TTLO between MEC servers, what is expected is not only to save computing resources, but also to achieve load balancing goal with fewer task transferred. According to the characteristics of the MEC network shown in Figure 4a, there is only one effective load transfer method that can both make the estimated load of two MEC servers tend to the *mean load line* together and minimize the TTLO in a task transfer, i.e., the load transferred from RHL-MEC to RLL-MEC. Naturally, there are also two other methods of load transfer, i.e., from one RHL-MEC to another RHL-MEC and from one RLL-MEC to another RLL-MEC. However, these two methods are invalid, since they cannot balance the network load but increase the TTLO in vain.

Through the above discussion, we only need to transfer the load of RHL-MEC above the *mean load line* to RLL-MEC. Nevertheless, it is only an ideal situation where all users covered by the SBS are receiving task processing service in Figure 4a. However, sometimes, this is not the case. The actual load, which includes the load of the task being processed and the load of the requested task, may be above or below the *mean load line*. As shown in Figure 5, only when the actual load is above the *mean load line*, and only the part of the load that exceeding the *mean load line* needs to be transmitted; otherwise it does not.

In each decision slot, the UEs covered by each SBS may apply for task processing. We let the set of RHL-MECs whose actual load above the *mean load line* be denoted as M⊂B and the set of RLL-MECs be denoted as S⊂B. Furthermore, it is obvious that M∪S⊆B and M∩S=∅. We let the task that is applied at that moment be represented as (n,m), which indicates that it belongs to $n_{th}$ UE $\in U_{m}$ of m${}_{th}$ SBS ∈B. Then, we let the set of all *i* tasks that are applied from M at that slot be denoted as $A=\{\left(n_{1},m_{1}\right),\left(n_{2},m_{2}\right),...,\left(n_{i},m_{i}\right)\}$. Each task (n,m)∈A gets a task offloading instruction through the load balancing decision, which is represented as $x_{n,m}\in S$. Then, the set of offloading instructions of all tasks (n,m)∈A can be represented as $X=\{x_{n_{1},m_{1}},x_{n_{2},m_{2}},...,x_{n_{i},m_{i}}\}$. Furthermore, the task offloading instruction $x_{n,m}$ of the task (n,m)∉A is equal to *m*, which means that the task will be offloaded in IMEC.

We define a function h(a,b), which is represented as follows:(9)$$h\left(a,b\right)=\left\{\begin{array}{cc} 1, & \;a\;=\;b,\;\\ 0, & \;\mathrm{otherwise}.\; \end{array}\right.$$

Then, the load $l_{n,m}$ that is brought to an MEC server by task (n,m), through the offloading instruction $x_{n,m}$, can be represented as follows:(10)$$l_{n,m}=l_{n,m}^{I}\cdot h\left(x_{n,m},m\right)+l_{n,m}^{N}\cdot \left(1-h\left(x_{n,m},m\right)\right).$$
where $h(x_{n,m},m)$ indicates the position where the task will be offloaded. If $h(x_{n,m},m)=1$, it indicates that the task will be offloaded in IMEC, then $l_{n,m}=l_{n,m}^{I}$. Furthermore, if $h(x_{n,m},m)=0$, it indicates that the task will be offloaded in NMEC, then $l_{n,m}=l_{n,m}^{N}$.

The decision benefit function, which indicates the quality of load balancing, can be expressed as follows:(11)$$f=\sum_{\alpha \in (M\cup S)}\left|\left(\sum_{(n,m)\in A}l_{n,m}\cdot h\left(x_{n,m},\alpha \right)+L_{\alpha }^{current}\right)-L^{mean}\right|,$$
where $L_{\alpha }^{current}$ is the load currently being used and $L^{mean}$ represents the value of *mean load line*. $h(x_{n,m},\alpha )$ indicates whether the task is offloaded in the MEC server of $\alpha_{th}$ SBS. Then, function *f* represents the sum of the distance from the load of these MEC servers in M∪S to the *mean load line*.

From Equation (Equation 11), the decision benefit function of a subtask can be obtained, which can be expressed as follows:(12)$$f\left(t\right)=\sum_{\alpha \in (M\cup S)}\left|\left[\sum_{(n,m)\in A}l_{n,m}\cdot h\left(x_{n,m},\alpha \right)+L_{\alpha }^{current}\left(t\right)\right]-L^{mean}\right|,$$
where *t* is the time when the subtask starts to execute. Then $L_{\alpha }^{current}\left(t\right)$ represents the load of the MEC server of $\alpha_{th}$ SBS at time t on the timeline.

The optimization goal in this paper is to minimize f as expressed as follows:(13)$$\min_{X}f.$$

### 4.3. Solving Method

Due to the differences of tasks embodied in T, the processing load and TTLO of each task may vary. It is likely to happen that a task with smaller processing load may cause a larger TTLO than other tasks or vice versa. Therefore, the optimization problem as shown in function (Equation 13) is NP-hard. In this section, we propose an efficient load balancing scheme to reduce the complexity of solving the problem and consequently shorten the running time. Afterwards, we compare our algorithm with the genetic algorithm (GA), which is one of the solutions to solve the NP-hard problem.

#### 4.3.1. Load Balancing Algorithm

With Equations (Equation 3) and (Equation 8), load allocation of the IMEC to complete the task within the minimum delay requirement and additional load overhead caused by the task transmission delay can be obtained, respectively. The two parameters affect the task offloading decision, jointly. In order to simplify the problem, we propose a new parameter of the task (n,m) to establish the relationship between them, which is defined as *task unit load transfer overhead* (TULTO), expressed as follows:(14)$$o_{n,m}=\frac{l_{n,m}^{Extra}}{l_{n,m}^{I}},$$
where $o_{n,m}$ indicates how much additional load overhead will be caused by a unit load transfer of the task. It can directly reflect the transfer quality of the task, since TTLO decreases as $o_{n,m}$ goes down when transferring the same load.

MEC is committed to reducing the computation delay of task offloading strategy algorithm to meet the delay requirements of users. Therefore, the computational complexity of load balancing algorithm must be as low as possible. Based on TULTO, a load balancing algorithm is proposed as shown in Algorithm 1, which has low complexity and a stable and effective load balancing effect. The detailed process of the algorithm is shown as follows.
**Algorithm 1** Load Balancing Algorithm1:Input: Current estimated load and actual load of each MEC server, set of the tasks applied at that moment and the task information set T of them;2:Output: Set of task offloading instruction X;3:$L^{mean}$ is obtained from the estimated load;4:According to $L^{mean}$, M and S are obtained from the actual load of each MEC server;5:A is obtained according to M;6:**for** each task (n,m)∉A
**do**7: Set the task offloading instruction $x_{n,m}=m$;8:**end for**9:Calculate $o_{n,m}$ for each task (n,m)∈A;10:Calculate $L_{\beta }^{capacity}$ for each MEC server in the set S;11:**while**S≠∅**do**12: **if**
$L_{\alpha }^{actual}>L^{mean}$, where the $\alpha_{th}$ MEC server has the highest actual $L_{\alpha }^{actual}$ in the set M
**then**13:  Select the task (n,m)∈A with the smallest $o_{n,m}$ from the $\alpha_{th}$ MEC server;14: **else**15:  break16: **end if**17: Select the $\beta_{th}$ MEC server with the lowest actual load $L_{\beta }^{actual}$ in the S;18: **if**
$l_{n,m}^{N}\leq L_{\beta }^{capacity}$
**then**19:  $x_{n,m}=\beta$;20:  $L_{\beta }^{capacity}=L_{\beta }^{capacity}-l_{n,m}^{N}$;21:  $L_{\beta }^{actual}=L_{\beta }^{actual}+l_{n,m}^{N}$;22:  $L_{\alpha }^{actual}=L_{\alpha }^{actual}-l_{n,m}^{I}$;23:  Remove task (n, m) from A;24: **else**25:  Remove β from S;26: **end if**27: Append $x_{n,m}$ to X;28:**end while**29:**if**A≠∅**then**30: Set the task offloading instructions of all the tasks in A, $x_{n,m}=m$;31:**end if**32:**return**X.

*Step 1:* In each time slot, each SBS estimates the load of users under its coverage so that the value of $L^{mean}$ of *mean load line* is obtained. With $L^{mean}$, the set of RLL-MECs can be obtained and denoted as S. The actual load of an MEC server is expressed as follows:(15)$$L^{actual}=L^{current}+L^{apply},$$
where $L^{current}$ is the load of the task being processed and $L^{apply}$ is the load of the requested task at that slot. The set M is composed of the RHL-MECs whose the actual load $L^{actual}$ is higher than $L^{mean}$.

*Step 2:* For the tasks requested by this slot, if the task (n,m) comes from an MEC server that does not belong to M, it will be processed in IMEC, which indicates that the task offloading instruction value $X_{n,m}$ is equal to *m*.

*Step 3:* For all the tasks (n,m) from the MEC servers in M, the TULTO $o_{n,m}$ of them is calculated. For each MEC server in S, the load capacity $L^{capacity}$ of it is calculated. As is shown in Figure 5, $L^{capacity}$ represents how much load can be transferred to an RLL-MEC to maintain long-term load balancing, since not all the UEs of the RLL-MEC may apply for tasks at the same time. It is necessary to prevent excessive load from being transmitted into the RLL-MEC.

*Step 4:* The key of the load balancing algorithm is such that, at each iteration of the algorithm, it selects the task with the lowest TULTO of the RHL-MEC, whose actual load is the highest in M, and transfers it to the RLL-MEC with the lowest actual load in S. Thus, the value of $X_{n,m}$ of the task equals the index of the RLL-MEC. Then, the parameters are updated, including the load capacity of the RLL-MEC and the actual load of the RHL-MEC and the RLL-MEC. It is worth noting that load transfer is performed only when the load of the RHL-MEC is higher than $L^{mean}$. Otherwise, the iteration will come to an end. Furthermore, if the load capacity of an RLL-MEC is lower than the load $l_{n,m}^{N}$ of the task, it indicates that the task cannot be offloaded in this RLL-MEC. Otherwise, a certain degree of load instability will happen, since the actual load of the RLL-MEC may exceed $L^{mean}$ at some time. In this case, we remove it from S before S is empty to exit the iteration.

*Step 5:* Due to the existence of TTLO, after the iteration, there are still some tasks in A that may not get the task offloading decision instructions, and these tasks should be executed in their IMEC.

#### 4.3.2. GA Approach

To solve the NP-hard problem in the MEC network with GA, some adjustments need to be made to fit the model, including gene type, chromosome length, and fitness function.

*Chromosome length:* For a task, a gene on a chromosome is used to represent the task offloading instruction. In a decision slot, since all the tasks in A need to be determined with respect to which MEC it is to be executed on, the chromosome length is supposed to correspond to the number of tasks in set A.

*Gene type:* the value of a gene indicates the specific MEC server in which the task is offloaded. There are two options for the task in set A. One is to offload in IMEC, and the other is to be offloaded in the MEC server in S. Therefore, the gene type contains all the elements in S and the index of its IMEC.

*Fitness function:* for a chromosome, the fitness value describes the quality of the decision to offload these tasks in A. Equation (Equation 11) is used as the fitness function. The smaller the fitness value is, the better the fitness of the chromosome is. Therefore, the goal is to find the chromosome with the smallest fitness value during the GA iteration. The specific process of the GA is as shown in Algorithm 2.
**Algorithm 2** GA Approach1:Input:A, M, S;2:Output: the chromosome with the smallest fitness value;3:Initialization parameters: population size, maximum number of iterations, crossover probability, mutation probability;4:Obtain the gene type of each gene in a chromosome according to S;5:Obtain the chromosome length according to A;6:Generate initial population according to the gene type, chromosome length, and population size;7:**while** current iteration number < maximum number of iterations **do**8: Calculate the fitness value of each chromosome;9: Update the chromosome with smallest fitness value;10: **if** current smallest fitness value < a certain value **then**11:  break12: **end if**13: Population selection by roulette;14: Population crossover according to the crossover probability;15: Population mutation according to the mutation probability;16:**end while**17:**return** the chromosome with the smallest fitness value.

#### 4.3.3. Time Complexity Analysis

(1) *Load balancing algorithm*:

The number of iterations of the algorithm is mainly related to the number of tasks *n* in A. The execution of the algorithm mainly consists of two parts, one is to sort the *o* (TULTO) of these n tasks, and the other is to offload them.

By using the quicksort algorithm, the sorting time is approximately O(nlogn). To offload each task, some parameters updating operation need to be conducted, and the time spent is approximately O(n). Therefore, the overall time complexity is O(nlogn).

(2) *GA*:

Supposing that the iteration number of GA is set to *i*, the population size is set to *p*, and the chromosome length is set to *n*. Each iteration mainly consists of four parts. The calculation of fitness value and mutation are related to *n* and *p* jointly, and the time complexity is O(pn). Selection and crossover are related to *p*, and the time complexity is O(p). Then, the time of the entire iteration is approximately ip(2+2n). Therefore, the time complexity of GA is O(ipn).

## 5. Simulation Analysis

This section describes how simulation experiments were conducted to prove the effectiveness of our solutions, including load estimation by user load prediction, load timeline for subtask, and the load balancing algorithm. The whole experimental environment and the experimental process were simulated with *python*, running on a laptop with Intel Core i7-6650U 2.20 GHz processor and 16 GB RAM.

### 5.1. Algorithm Performance and Iterative Process

In this experimental scenario, the number of SBSs under the coverage of MBS in a community is 8, and the number of UEs covered by each SBS is set to a random number ranging from 1 to 50. The transmission rate of optical fiber is set to 1 Gbps. Each user has a delay-sensitive task to process, the size of which ranges from 1 to 10 Mbit. Meanwhile, the number of CPU cycles required to complete the task ranges from 500 to 1000 M, and the minimum delay requirement for the task ranges from 50 to 200 ms. The above three parameters are randomly set within their ranges, and we let all UEs applied for tasks at the same time. After that, our algorithm is used to balance the load. It is worth noting that the load estimation is mainly for UEs without offloading tasks. However, in this experiment, all UEs have applied for task processing, so there is no prediction load for UEs without offloading tasks. In other words, the estimated load of the MEC server is exactly equal to its actual load at that moment.

Figure 6a shows the actual load of each MEC server before load balancing, and the red line represents the load value of the *mean load line* at this decision. Then, through load balancing, the load of each MEC server is shown in Figure 6b, in which it can be seen that they are all closer to the *mean load line*. Furthermore, in Figure 6c, the change process of the load of each MEC in the entire algorithm iteration is shown. Furthermore, we can see that, in each iteration, the task is transferred continuously from the MEC server with the highest actual load to the one with the lowest actual load until each server approaches the *mean load line*.

### 5.2. Reduction of The Impact of the Ping-Pong Effect

In this experimental scenario, the simulation parameters are kept the same as those in Section 5.1. Nevertheless, the tasks of all UEs are not applied at the same time. Accordingly, the decision slot interval is set to 10 ms, and each UE that has not yet applied for a task has a 50% probability of applying for a task in each decision slot. The simulation time is set to 10,000 ms, and the number of decision epochs is set to 1000.

In each decision slot, for the requested task, our load balancing algorithm with or without load estimation is used, respectively, to balance the load. It is worth noting that, according to the characteristics of the load balancing algorithm, the algorithm will stop after a certain number of iterations, which depends on the number of tasks applied in the decision slot. Furthermore, it is set to stop automatically in each decision slot. Furthermore, load balancing decisions without load estimation depends on the actual load of each MEC server, which will affect the value of $L^{mean}$, and then the ping-pong effect is aggravated in loading balancing.

Figure 7a,b shows the load change of each MEC server with or without load estimation over time, respectively. It can be seen that both of them can keep the load of those MEC servers near the average. However, as shown in Figure 7c,d, without load estimation, the number of tasks transferred is increased accordingly, which causes the growth of TTLO. Furthermore, these are exactly the impacts of the ping-pong effect. The impact can be reduced effectively through load estimation, since the approximate load of MEC servers rather than the current actual load is considered in the proposed algorithm, which will lessen the influence of the highly dynamic environment in the MEC network.

### 5.3. Systematic Stability with Load Estimation

In this experimental scenario, the number of MEC servers is set to 6, and the load of each server is a random number ranging from 0 to 500 before load balancing. Firstly, each MEC server is set with full load. Then, a certain MEC server in a decision slot is randomly selected to drop its load to 0 instantly and recover its load in the next decision slot, which can be considered as a load fluctuation. By using the load balancing algorithm with or without load estimation, respectively, the inflow load, outflow load, and actual used load of the MEC server can be obtained and compared, respectively, in the entire process.

The inflow load includes the load of applied tasks and the load transferred from other MEC servers. The outflow load refers to the load transferred from this MEC server to another. Furthermore, the actual used load refers to the load actually used to process the tasks. In fact, the actual used load is equal to the inflow load minus the outflow load. The inflow load and the outflow load can reflect how the scheme works to balance the load.

In this experiment, *MEC2*, *MEC3*, and *MEC6* are RHL-MECs, and *MEC1*, *MEC4*, and *MEC5* are RLL-MECs. Firstly, as shown in Figure 8a,c, the inflow load of RLL-MECs keeps at a fixed value all the time, namely the value of the *mean load line* when there is no load fluctuation. In Figure 8a, when the fluctuating MEC server is RHL-MEC; e.g., when the decision epoch equals to 10, the inflow load of *MEC2* which is the fluctuating MEC server drops to 0, and the inflow load of the all RLL-MECs is affected. When the fluctuating MEC server is RLL-MEC, e.g., when the decision epoch equals 20, the inflow load of *MEC5*, which is the fluctuating MEC server, drops a little, but not to 0, since part of RHL-MECs’s load flows into it. In this case, only *MEC5’s* inflow load is affected. In Figure 8c, no matter whether the fluctuating MEC server is RHL-MEC or RLL-MEC, the inflow load of all MEC servers is affected.

Secondly, as shown in Figure 8b,d, when there is no load fluctuation, only the RHL-MECs have outflow load, which is equal to the part of the inflow load that is higher than the *mean load line*, and the outflow load of the RLL-MECs is 0. In Figure 8b, only when the fluctuating MEC server is RHL-MEC does its outflow load drop to 0. No other MEC server’s outflow load is affected. In Figure 8d, similarly, no matter whether the fluctuating MEC server is RHL-MEC or RLL-MEC, the outflow load of all MEC servers is affected.

Thirdly, in Figure 8e,f, when there is no load fluctuation, the actual used load of all MEC servers is at the *mean load line*. The change of Figure 8e is similar to that of Figure 8a. Furthermore, in Figure 8f, the actual used load of all MEC servers is affected, and their values are equal when load fluctuation occurs.

Generally speaking, when the load of an MEC server fluctuates, load estimation can reduce the number of affected MEC servers in terms of the three types of load through load balancing. When the fluctuating MEC server is different, the system may be affected differently. Specifically, when the fluctuating MEC server is RHL-MEC, the inflow load and actual used load of these RLL-MECs will be affected, but only the outflow load of the RHL-MEC with load fluctuation will be affected. Furthermore, when the fluctuating MEC server is RLL-MEC, only the inflow load and the actual used load of other RLL-MECs are affected. Without load estimation, the load fluctuation of the fluctuating MEC server can be slowed down, which is to average the load fluctuation to other MEC servers. However, all three types of load of all MEC servers will be affected. Hence, with load estimation, the impact of load fluctuation can be reduced and the stability of the load of the system can be maintained to a certain extent.

### 5.4. Systematic Stability with Load Timeline for Subtask

In this experimental scenario, the number of MEC servers is set to 8, and the number of UEs of each MEC server is set to a random number ranging from 1 to 30. Only sequential subtasks are discussed here, because the sequential-parallel-hybrid subtasks model can be considered as a sequential subtasks model in a hierarchical way. Each UE has three sequential subtasks, and the number of CPU cycles required to complete each subtask is a random number ranging from 400 to 800 M. The minimum delay requirement of each subtask is fixed at 20 ms. Then, the load balancing algorithm with or without load timeline is used, respectively, for the load balancing of the subtasks.

If there is a load timeline, the load balancing of subtasks depends on the corresponding load of each MEC server, when the execution of subtasks starts on the timeline. Otherwise, it depends on the load of each MEC server at the current time.

As shown in Figure 9, the introduction of load timeline for subtasks can make it more concentrated around the average value for the load of each MEC server through load balancing. Otherwise, a certain load fluctuation may appear. Because load estimation provides each subtask with the load status at the beginning of its execution, accordingly, a more accurate and reasonable load balancing decision will be made.

### 5.5. Comparision with GA

In this experimental scenario, GA is further used for load balancing on the basis of the simulation experiment in Section 5.1 and then compared with our load balancing algorithm (LBA) in terms of the time consumption. The crossover probability of GA is set to 0.6 and the mutation probability is set to 0.01. GA is made to stop iterating when its balancing error reaches that of LBA, which is represented by *f* in Equation (Equation 11).

As shown in Figure 10, the population size of GA is set to 20, 35, 50, 100, and 200, respectively, and the experiment is carried out 50 times. Due to the random search of GA, the search time fluctuates with the same population size. As shown in Table 1, as the population size increases from 20 to 100, the average time consumed by GA increases, but the average generation decreases. The reason is that with smaller population size and search scope, more crossover and mutation are needed to expand the search scope so as to search for a better solution. Nevertheless, the larger the population size is, the more the amount of computation of each generation will increase, and thus the more the total search time increases. Therefore, it is necessary to set a suitable population size of GA according to specific situation to get better performance. According to Figure 10 and Table 1, compared with GA, LBA has better performance in terms of time consumption in the model in this paper.

In order to further compare the load balancing performance of the two algorithms, the condition for GA to stop iteration is set as when the number of iteration reaches a certain value, and the simulation parameters after the iteration can be obtained. The larger the population size is, the higher the stability of GA is. Hence, the population size and generation are both set to 200 according to Table 1.

As shown in Figure 11a, the number of tasks transferred by the two algorithms are similar, because the amount of load to be balanced is fixed and the difference between tasks is not particularly large; moreover, the TTLO is relatively small for the load of a task. However, in Figure 11b, the TTLO of LBA is significantly lower than that of GA, because it is based on TULTO, which makes it tend to transfer the tasks that are relatively less burdensome for the entire system. As shown in Figure 11c, with GA, the error varies every time through the load balancing, and sometimes a better solution is found than LBA. Nonetheless, there is a certain amount of randomness and instability existing in GA, and the iteration time of GA is much longer than LBA. It is very clear that the execution time of load balancing decisions will affect the size of resource allocation. Obviously, if it is too time-consuming, additional task-computing resources need to be allocated to make up for time consumption.

### 5.6. Scalability Analysis of Algorithm

Due to the highly dynamic environment in the actual case, it is necessary for the load balancing algorithm to have a certain degree of scalability. For this purpose, the number of SBSs and the number of UEs under each SBS are changed, respectively. Furthermore, the time consumption and load balancing effect of LBA and GA can thus be attained.

Specifically, in Figure 12a,b, the number of UEs of each SBS is fixed at 50, and the number of SBSs is gradually increased from 4 to 12. The population size of GA is set to 50, and the generation is fixed at 500 according to Table 1. In Figure 12a, we can see that with the increase in SBS, the time consumption of LBA is always at a low level, obviously contrary to that of GA increases. In Figure 12b, there is no obvious change for balancing error of LBA. Furthermore, the load balancing effect of GA becomes worse with the increase in SBSs, since the larger the balancing error is, the worse the load balancing effect is.

In Figure 12c,d, the number of SBSs is fixed at 8, and the number of UEs of each SBS is gradually increased from 10 to 100. Furthermore, the trend of these two lines is similar to that of Figure 12a,b. Therefore, LBA has better scalability than GA in terms of algorithm complexity and load balancing effect. In order to show the scalability of LBA in more detail, Figure 12e,f are given in addition.

In Figure 12e, the time consumption increases with the rise of the number of SBSs and UEs, but it is always at a lower level compared with GA. In Figure 12f, on the contrary, the balancing error is reduce with the rise in the number of SBSs and UEs. The reason is that, unlike the constant increase in total load, the balancing error is almost at the same size, which makes the proportion of the balancing error decrease. In general, LBA has a good scalability performance.

## 6. Conclusions

In this paper, the issues of mobile edge computing with load balancing were studied in the ultra-dense network. Firstly, as for the ping-pong effect existing in load balancing, the reason for its formation and its generation process are explained in detail. Then, in order to reduce the cost and burden brought to the entire network by the ping-pong effect, the main idea of SDN was introduced into the system framework. Furthermore, on the basis of that, a load balancing algorithm based on load estimation was later proposed. In our algorithm, by introducing the new concept of *task unit load transfer overhead*, an effective solution to the NP-hard problem was quickly obtained in task offloading. Moreover, the load timeline of the MEC server and the hierarchical method of subtasks for divisible tasks were proposed to solve their instability in load balancing. Finally, simulation results also demonstrate the effectiveness of load estimation in reducing the impact of the ping-pong effect, as well as the efficiency and rapidity of our algorithm compared with GA. However, for the differences between theoretical research and practice, there are many other factors affecting the results that have not taken into account. Therefore, in the follow-up research work, solutions for more complex scenarios will be considered.

## Figures and Tables

**Figure 1 sensors-21-03135-f001:**
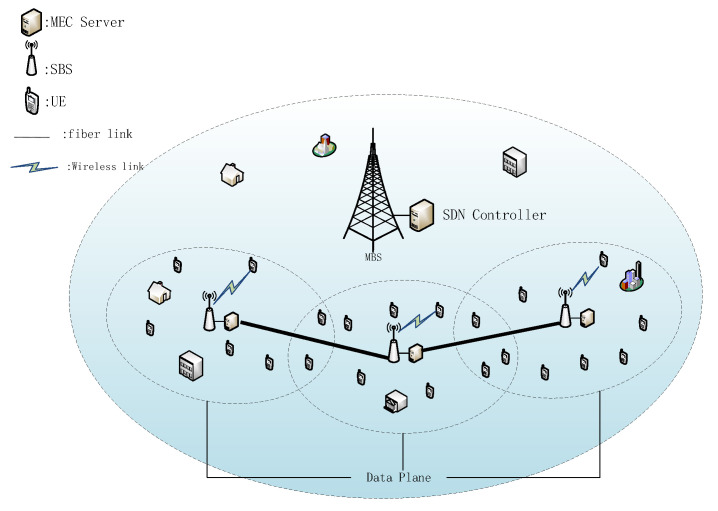
System framework.

**Figure 2 sensors-21-03135-f002:**
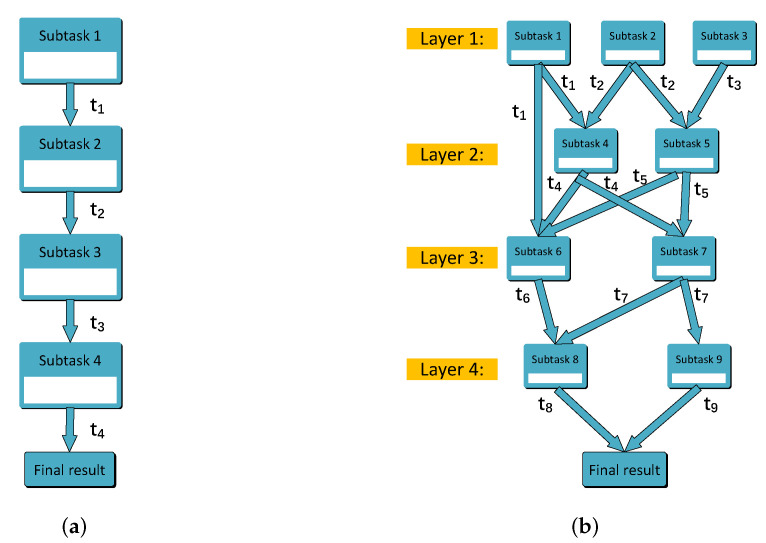
Subtask model. (**a**) Sequential subtasks model; (**b**) sequential-parallel-hybrid subtasks model.

**Figure 3 sensors-21-03135-f003:**
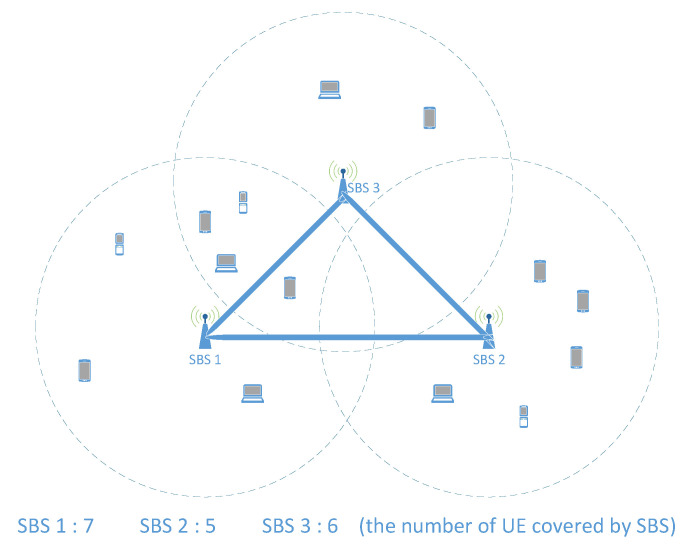
Illustration of a case of MEC in UDN.

**Figure 4 sensors-21-03135-f004:**
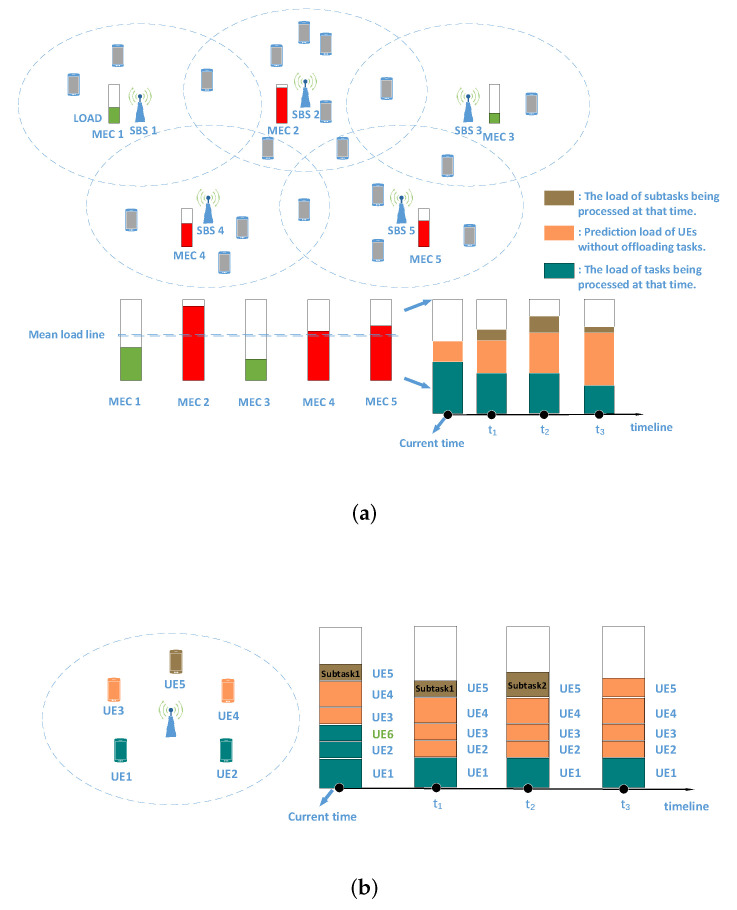
Illustration of load estimation. (**a**) Global load estimation; (**b**) details of load estimation and timeline.

**Figure 5 sensors-21-03135-f005:**
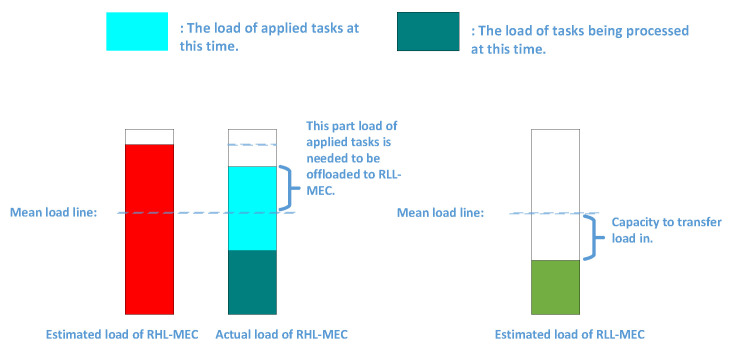
Illustration of how to offload tasks for an RHL-MEC and the load capacity of an RLL-MEC.

**Figure 6 sensors-21-03135-f006:**
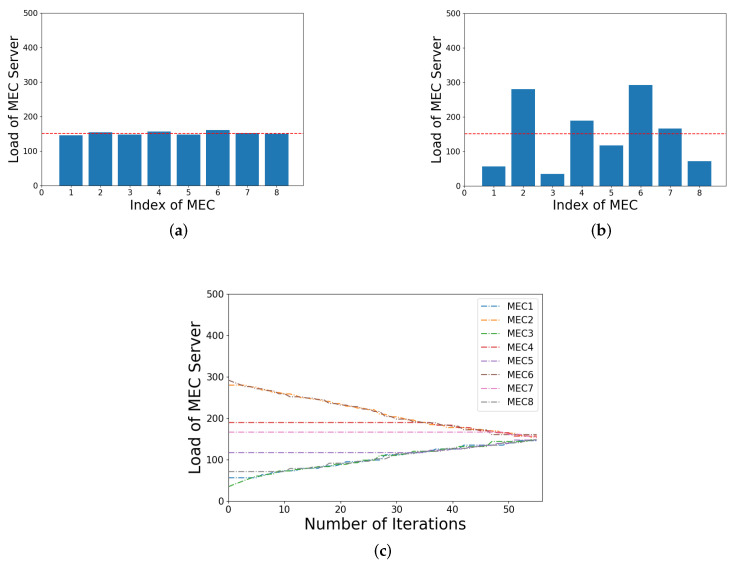
Illustration of the load of each MEC before and after load balancing and its algorithm iteration process: (**a**) before load balancing; (**b**) after load balancing; (**c**) iterative process.

**Figure 7 sensors-21-03135-f007:**
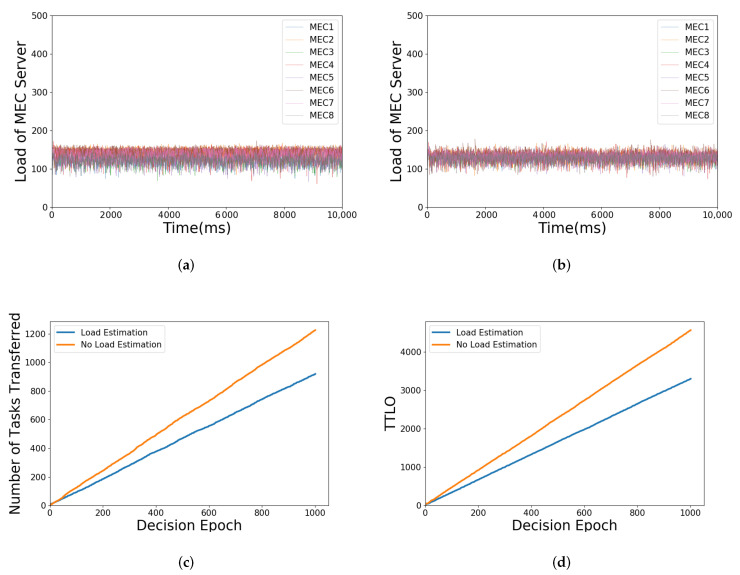
Illustration of simulation parameters under load estimation or not: (**a**) load balancing with load estimation; (**b**) load balancing without load estimation; (**c**) number of tasks transferred when load estimation or not; (**d**) TTLO when load estimation or not.

**Figure 8 sensors-21-03135-f008:**
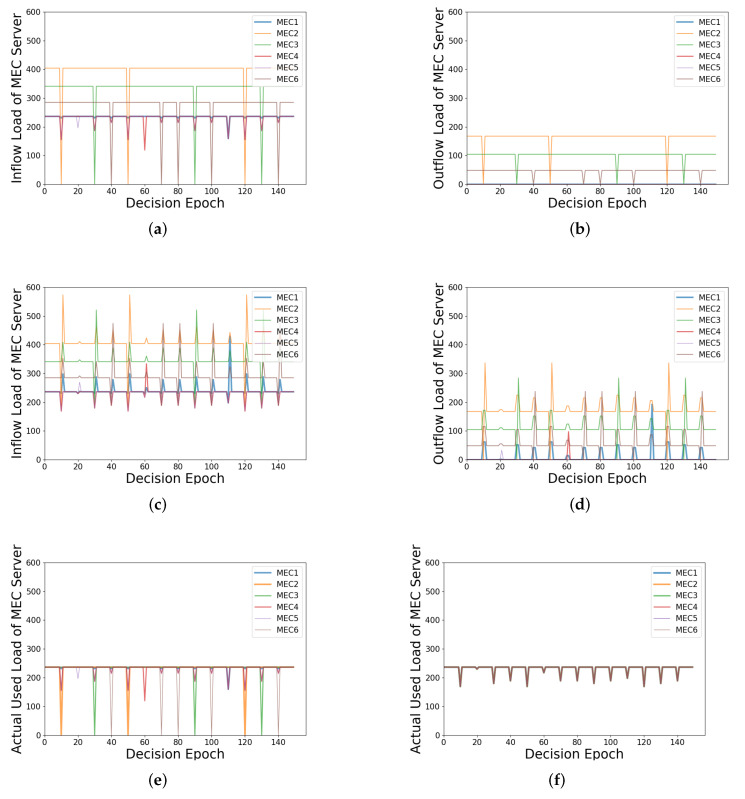
Illustration of various loads of the MEC server under load estimation or not: (**a**) inflow load with load estimation; (**b**) outflow load with load estimation; (**c**) inflow load without load estimation; (**d**) outflow load without load estimation; (**e**) actual used load with load estimation; (**f**) actual used load without load estimation.

**Figure 9 sensors-21-03135-f009:**
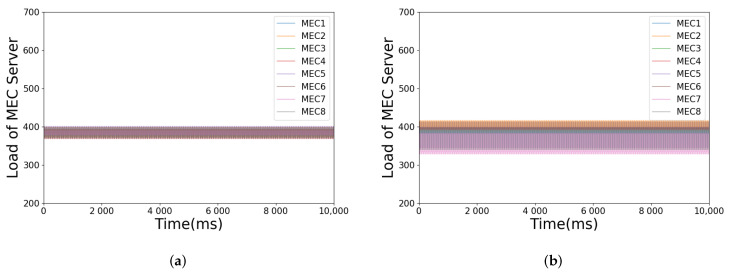
Illustration of whether the load balancing of subtasks is based on the load timeline: (**a**) load balancing with timeline for subtask; (**b**) load balancing without timeline for subtask.

**Figure 10 sensors-21-03135-f010:**
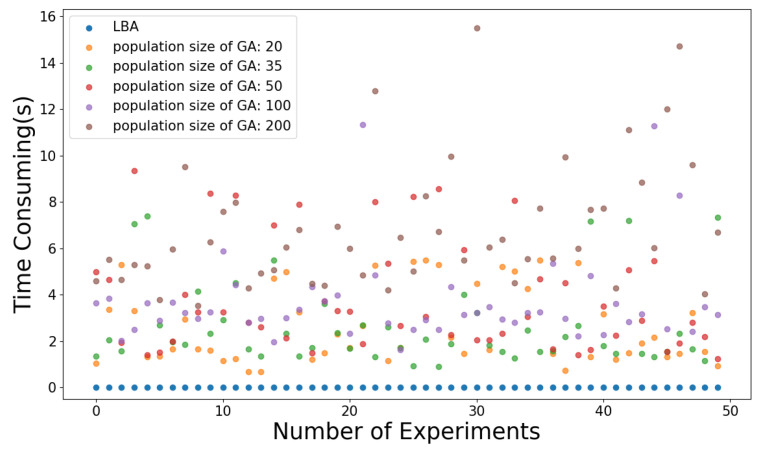
Illustration of time consumed by the algorithm.

**Figure 11 sensors-21-03135-f011:**
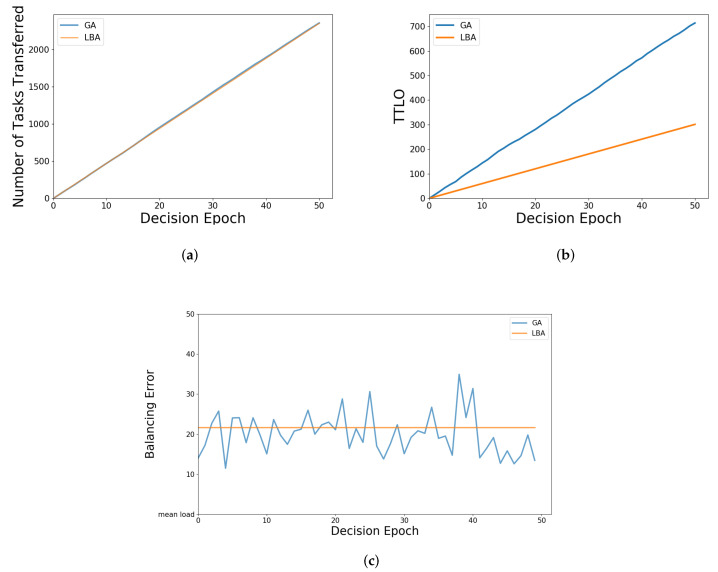
Illustration of simulation parameters of LBA and GA: (**a**) number of tasks transferred of LBA and GA; (**b**) TTLO of LBA and GA; (**c**) balancing error of LBA and GA.

**Figure 12 sensors-21-03135-f012:**
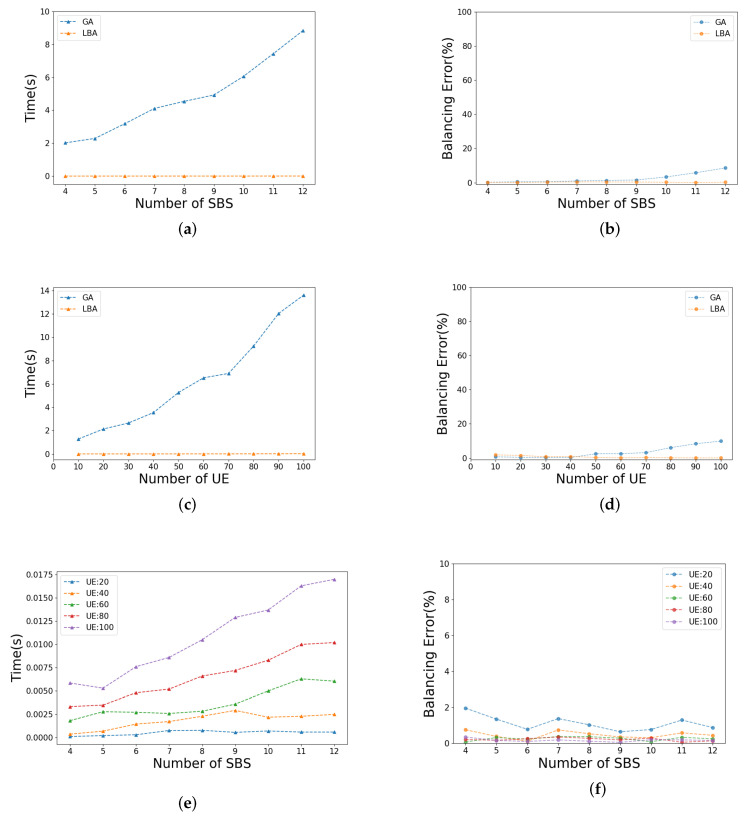
Illustration of scalability analysis: (**a**) time consumption of GA and LBA as the number of SBS increases; (**b**) balacing error of GA and LBA as the number of SBS increases; (**c**) time consumption of GA and LBA as the number of UE increases; (**d**) balacing error of GA and LBA as the number of UE increases; (**e**) time consumption of LBA under different numbers of UE; (**f**) balacing error of LBA under different numbers of UE.

**Table 1 sensors-21-03135-t001:** Average time consumed by algorithm and average generations of GA.

	LBA	GA:20	GA:35	GA:50	GA:100	GA:200
time (s)	0.0012	3.239	3.275	3.371	4.445	6.428
generations	-	1220	680	456	284	184

## Data Availability

Not applicable.

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
