# Peer review of "Enhancing Mobile Edge Computing with Efficient Load Balancing Using Load Estimation in Ultra-Dense Network"

_sensors, 2021, doi:10.3390/s21093135_

Round 1

Reviewer 1 Report

SDN is understood by many readers as a routing technique, often based on OpenFlow. For that reason, I propose to state in the abstract, as well as in the introduction, that SDN paradigm is applied to the task processing allocation (thus routing their corresponding messages in consequence). 
Here are some of the imperfections I noticed:
Line 141: An orphaned title of a chapter. 
Lines 203 and 206: A lack of space before “offloading”.
Line 274 and 275 and 280: The given numbers of UEs in SBS1 and SBS2 do not correspond to those depicted in Fig. 3.
Line 304: “e.g” starts a sentence. Use a capital letter.

Reviewer 2 Report

The authors should carefully proofread the manuscript from beginning to end. Reading of the text is hard at some points.

Software Defined Networking is introduced in section 3.1 as a required technology for implementing the proposal. In lines 154-164, authors explain the terms “control plane” and “data plane”, but these definitions do not adjust to SDN architecture. In SDN, the data plane (network elements) is the network layer that forwards the traffic based on the configuration supplied by the control plane. The sentence “data plane getS the global view of the network which includes user distribution, MEC server load status, and task request information, etc” is too general and imprecise.

Line 193 says “we do not take the process of task uploading into account in our model”. However, that is not totally true because transmission from IMEC to NMEC is consider in equation [5]. Why the uploading from UE to IMEC is not considered, but however, transmission from IMEC to NMEC is considered?

In my opinion, in section 3.3.2, the use of terms “sensing” and “prediction” is not adequate. For example, in line 293, it is said “user local prediction can be used to sense the load of a MEC server”.  Sensing is synonym of observing and noticing; and prediction means “to say what will happen in the future”. If so, the sentence should say “user local sense can be used to predict the load of a MEC”. That also happens in other points of the paper.

Lines 264-270: The definitions of RHL-MEC and RLL-MEC are not clear. If the election of RHL-MEC and RLL-MEC supposes that all the UEs offload their tasks (I suppose that the same number of tasks, and tasks demanding the same requirements) at the same time”, why are not defining as the MEC of the SBS with less UEs and the MEC of the SBS with more UEs?

In SDN is used to obtain accurate information about “user distribution, MEC server load status, and task request information”, why case 1 and case 2 situation could occur? Ping-pong problem should be described in a clearer way. Time reference is not considered, and it could be useful to describe the problem.

Line 274-276: “all the six UEs of SBS2” -> in the figure, there are only 5 UEs in SBS2. “while SBS1 has less than this number”-> This is not seen in the figure. In the figure, SBS1 has 7 UEs. However, the figure does not show how many tasks are offloaded. Again, lines 289-2191 refer to a situation that is not represented in figure 3.

“Timeline” is a key aspect named in line 344. But it is not defined enough.

Figure 4: label “prediction load of UEs without offloading tasks”? what does it mean? In the text is said “predicted load of the UEs which has not offloaded the tasks yet”. In my opinion, both definitions are not the same. Also in figure 4, the label of color brown is “the load of subtasks that be processed at moment t”… which is moment t? Figure 4, the definitions and its explanation should be improved. “Timeline” is a key aspect named in line 344. But it is not defined enough.

Figure 4: label “prediction load of UEs without offloading tasks”? what does it mean? In the text is said “predicted load of the UEs which has not offloaded the tasks yet”. In my opinion, both definitions are not the same. Also in figure 4, the label of color brown is “the load of subtasks that be processed at moment t”… which is moment t? Regarding figure 4,  the definitions and the figure explanation should be improved.

There is not any detail about the simulations. What simulator has been used? Is it an ad-hoc simulator?

Section 5.2. How many iterations are executed during each decision slot?

Line 519: The use of term “sensing” and “prediction” are confusing. “We use our load balancing algorithm with or without LOAD SENSING to balance the load”. It should be said “LOAD PREDICTION”. What prediction mechanism is used in the simulations?

Y-scale used in figures 7-a and 7-b is not adequate. Nothing can be seen clearly.

Section 5.3 should be improved. What happen in this experiment is not clear enough. How to identify a fluctuation MEC server in the figure? The figures 8-a to 8-e should be explained in more detail to support the conclusions.

Y-scale of figures 9-a and 9-b should also be modified.

Reviewer 3 Report

The manuscript is correctly presented. It addresses a relevant research topic framed in the land of Mobile Edge Computing (MEC) and Ultra-dense Networks (UDN). The proposal comes along with an experimental validation with methodological rigor and with a clear scope. Nevertheless, there are important improvements to be addressed before considering the publication of this research. They are summarized as follows:

  1. The relationship between this research and 5G networks is barely justified. For this reviewer, there are no particular insights to be noticed out of the scope of current LTE networks.
  2. The Task Model relies on three parameters (s,c,d). While it is true that s and d determination is straightforward, it is quite difficult to esimate the number of CPU cycles required to complete a task. This reviewer is skeptical on such determination unless a more elaborated reference on processing burden estimation in terms of CPU cycles is provided. Equation (1) describes (l) as an estimation variable, but the scope of "computing resources" is arguable.
  3. The experimental validation of both LBA and GA have been simulated. As there is a plethora of similar papers following such approach and due to significant differences arised from a simulated environment in comparisson with real implementation, the outcomes of this research in terms of efficient balancing is still arguable. First, there is no reference on the implementation platform/tool used to conduct the experiments. Second, altough splitting the control and data plane traffic in SDN is possible, high delay-sensitive scenarios like those reflected in this paper might perform considerably different in real implementations since, for instance, control-to-data-plane configuration delay, BS handover, or data-plane network performance metrics (delay, packet loss, throughput) play a role in the overall validation scheme. So, disregarding such conditions in simulated load-balancing schemens raises concerns towards their real contribution towards the state of the art.  
  4. I suggest reformulating the ideas drawn in the Conclusions section in the sake of clarity. Some phrases are simply paraphrasing the same ideas commented beforehand in the experiments.
  5. The manuscript writting must be considerably enhanced. It must undergo a extensive grammar revision. Spelling is to be checked as well. Example: Algorothm (p.20), whit (p.18), etc.

Round 2

Reviewer 2 Report

As I said in the first revision, in my opinion, the use of SDN term in this work is not adequate. SDN (Software Defined Networking) is a framework to control/define/manage the forwarding actions of network elements locating the control plane (decision tasks) out of the network elements (they just “forward” packets). Therefore, the concept of SDN is not adequate in this scenario. This work defines a centralized control of load-balancing decisions at the MBS based on information obtained from the distributed SBS and MEC servers. But this is not SDN.

Page 4, line 152 -> this paper presentS

Page 4, line 155 -> A large amount of UEs ARE

Page 10, line 363 -> In terms of the number of UEs

Page 13, line 485 -> to REDUCE

I’m sorry but I still believe that the use of terms “sense” and “prediction” is not appropriate. From your response I understand that the system uses the past behaviour of UEs to “predict” what would be the behaviour of the UEs which are not requesting for tasks. And this prediction is used to “estimate” the load of the MEC. Perhaps the use of “estimation” would be clearer.  In my opinion, the term "sense" is more linked to "detect", but you are estimating and not detecting or measuring the load.

The introduction of what you called “load sensing” is a key issue of this work. Therefore, I insist that the selection of an adequate terminology is required. In my opinion, the term "sense" is more linked to "detect", but you are estimating and not detecting or measuring the load.

Regarding section 5.1, I assume that “load sensing” is not being applied. From the text, a task is assigned to each UE (accordingly to the described random parameters). Therefore, there is no “prediction load of UEs without offloading tasks”. If it is right, it should be said more explicitly. 

Reviewer 3 Report

The fact that 5G and MEC are frequently mentioned together in the literature is not a solid statement to justify the suitability of the introduced proposal in the land of 5G networks, so removing 5G is quite more appropriate.

On the other hand, this reviewer still has important concerns with regard of the experimental validation. A rather basic "python-based simulation" of complex MEC scenarios has several limitations. In order to raise the results significance, a comparative assessment with similar simulation-based proposals should be included along with a deep results discussion.
